# When noise mitigates bias in human–algorithm decision-making: An agent-based model

Spencer Poodiack Parsons *, René Torenvlied

Department of Public Administration, University of Twente, Enschede, The Netherlands

☉ These authors contributed equally to this work.
* s.j.poodiack-parsons@utwente.nl

## Abstract

Algorithmic systems increasingly inform human decision-making in domains such as criminal justice, healthcare, and finance. Although algorithms can exhibit bias, they are much less prone to undesirable variability in judgments (noise) than human decision-makers. While presented as an advantageous feature of algorithmic advice, we actually know little about how (biased) algorithmic advice interacts with noisy human judgment. Does undesirable variability in human judgment decrease under noiseless algorithmic advice? Is bias in human judgment exacerbated or mitigated by noise in advice? To answer these questions, we built an agent-based model that simulates the judgment of decision-makers receiving guidance from a (more or less) biased algorithm or a (more or less) biased and noisy human advisor. The model simulations show that, contrary to expectations, noise can be desirable: human noise can mitigate the harms of algorithmic bias by dampening the influence of algorithmic advice. Noise in human advice leads decision-makers to rely more heavily on their prior beliefs, an emergent behavior with implications for belief updating. When decision-makers' prior beliefs are polarized, an asymmetry occurs: decision-makers respond only to interventionist advice and not to non-interventionist cues. Finally, the model simulations show that population-level variability in decision-making stems from occasion noise in the environment and not from noise in human advice. This result challenges the common wisdom that population-level noise can be straightforwardly decomposed into individual-level sources and questions the feasibility of noise audits in organizations. Together, these findings demonstrate that the absence of noise as a feature of algorithmic advice is not generally desirable, suggesting critical implications for how human-algorithm systems are designed, regulated, and evaluated.

## Introduction

Since the 1950s, research has shown that algorithms can outperform human decision-makers in tasks such as medical diagnosis and predicting criminal

**Data availability statement:** All simulation outputs, scripts used to reproduce the figures, and the full agent-based model code for this study are publicly available from the Open Science Framework (OSF) repository (https://osf.io/vzmd2).

**Funding:** The author(s) received no specific funding for this work.

**Competing interests:** The authors have declared that no competing interests exist.

recidivism [1]. A key advantage of algorithms is their ability to eliminate *noise* in decision-making, defined as "undesirable variability in judgments of the same problem" [2,3]. In their seminal work on *noise,* Kahneman et al. (2021) highlight how this problem arises in a wide variety of decision-making contexts, such as bail decisions and medical diagnosis. They note that wherever there is human judgment, there will be noise, which translates into system-level variability [2,4,5]. For example, when making a critical medical diagnosis, doctors are often found to disagree, even with themselves, on a second review of the same problem [2]. Kahneman et al. (2021) claim that algorithms reduce or even eliminate undesirable noise, though this claim has been criticized by others [2,6,7]. Yet, noise may not always be undesirable. In some contexts, such as collaboration games, a small amount of noise has been empirically found to improve performance [8].

While algorithms have the potential to eliminate or reduce noise compared to human decision-makers, a common challenge with algorithms is bias. Algorithmic bias occurs when algorithmic systems produce consistent systematic errors without a warranted rationale [9]. Algorithmic bias is often caused by 'dirty data,' in which the training data of an algorithmic system contains examples of discriminatory conduct [10]. Algorithmic bias is one reason that the full potential of algorithms to improve decision-making in contexts such as bail determinations has not been achieved [11]. Policymakers and scholars, therefore, have expressed serious concern about autonomous algorithmic decision-making in vital societal contexts, arguing that human professionals should have the final word in such decisions. However, algorithms acting as advisors could offer the alleged benefits of noiseless advice without legal, ethical, or practical objections to automated decision-making. But is noiseless advice always better? Noise influences human-algorithmic decision-making in complex ways. Additionally, we know little about how (biased) algorithmic advice interacts with human judgment. Does undesirable variability in human judgment decrease under noiseless algorithmic advice? Is bias in human judgment exacerbated or mitigated by noise in advice? There is limited understanding of these questions. This leaves practitioners and policymakers with little guidance about whether algorithmic advice would cause disproportionately adverse effects in a given context [3].

In the present study, we compare noiseless (algorithmic) advice to noisy (human) advice at various levels of bias [2,3,12]. Assuming algorithmic advice is noiseless, we can contrast it with noisy human advice. Importantly, human beings also have a range of biases that impact how they make decisions and respond to advice from either their colleagues or algorithmic systems. For example, scholars have found that police officers follow the advice of algorithmic systems primarily when they match their prior beliefs [13]. Likewise, healthcare professionals may hold (implicit) biases that impact how they evaluate algorithmic diagnostic advice [14,15]. Thus, human–algorithmic decision-making involves a complex interaction between noisy and biased humans and noiseless, though biased, algorithms. The present study offers new theoretical insight into the complex interaction between noise and bias in human-algorithmic decision-making. By building an agent-based model, we identify

conditions under which noiseless algorithmic advice brings more (or less) desirable effects on human judgment than noisy human advice.

In our agent-based model, decision-makers and subjects interact in a spatial environment, akin to police officers patrolling a neighborhood and interacting with residents, some of whom are 'bad' or 'guilty'. The decision-makers make random walks in a two-dimensional space and encounter randomly walking subjects (residents). These random spatial movements introduce occasion noise, generating variability in encounters between decision-makers and subjects [2,16]. Upon encountering a subject, the decision-maker forms a prior belief about the subject's likelihood of being 'bad' (e.g., guilty of a crime) based on observable characteristics. Subsequently, the decision-maker receives advice, either from a colleague or an algorithm. Based on that advice, the decision-maker adjusts their prior belief and decides whether to intervene by putting the resident in retention. Then the officer continues their patrol. We identify three population-level outcomes of decision-making in our model: *true positives* (a 'bad' resident in retention), *false positives* (a 'good' subject in retention), and *false negatives* (a 'bad' subject free to walk). To test the robustness of our results, we also specified a model in which subjects are lined up to pass decision-makers. That model reflects more controlled real-life contexts, such as immigration control of passengers at an airport.

Our simulations show that noise in advice does not increase variability in the population-level behavior of decision-makers. Instead, variability arises from environmental noise and specifically, the stochastic pattern of encounters between decision-makers and subjects. We also find that noise in human advice produces an emergent regularity: decision-makers tend to revert toward their prior beliefs. This effect is not a programmed confirmation bias but emerges from the interaction of uncertainty and stochasticity in advice. Thus, contrary to expectations, noise can be desirable: human noise can mitigate the harms of algorithmic bias by dampening the influence of algorithmic advice. Finally, the simulations reveal an asymmetry in populations with polarized priors: decision-makers respond to interventionist advice but largely ignore non-interventionist signals. Taken together, these contributions: (a) clarify the conditions under which algorithmic advice can be more (or less) harmful than human advice, (b) identify a mechanism through which noise mitigates bias, and (c) suggest testable predictions for experimental and field research. More broadly, the study demonstrates how agent-based modeling can illuminate trade-offs in human–algorithm decision-making across a wide variety of domains.

## Individual decision-making under advice

We first devised a formal model considering an individual decision-maker and biased advice from either an algorithm or a human (full model in Methods). Our individual-level model is informed by a classic formal model, which considers the impact of biased advice from advisors on politicians' decision-making [17,18]. Equation (1) is the foundation of our individual-level model (full model in Methods).

$$z_{id}^* = U(z_i) = z_{id}^{\beta_A} + \varepsilon_A \sim (-c, c), \text{ where } 0 \leq z_i^* \leq 1 \tag{1}$$

In this model, a decision-maker $i$ considers subject $s_d$ and holds a prior belief ($z_{id}$; $0 \leq z_{id} \leq 1$) about the subject's state. We model this prior belief as a draw from a random distribution of all possible prior beliefs about a subject (in Equation (1) initially as a uniform distribution). When the prior belief $z_{id} = 0$, the decision-maker is certain that the subject is not bad. When the prior belief $z_{id} = 1$, the decision-maker is certain that the subject is bad. The decision-maker is maximally uncertain about the subject's state when $z_{id} = 0.5$. The decision-maker updates their prior belief based on advice $A$ received. That advice can be biased with a value $\beta_A$. Advice can be *neutral* ($\beta_A = 1$), biased toward a more *interventionist* choice ($0 \leq \beta_A < 1$), or biased toward a more *non-interventionist* choice ($\beta_A > 1$). If, after receiving advice, the updated belief falls below the decision threshold, $z_{id}^* < 0.5$ the decision-maker will *not intervene*; otherwise, if $z_{id}^* \geq 0.5$, the decision-maker will *intervene*.

Ambiguous (human) advice comes with noise, which we model as a stochastic shock $\varepsilon_A$ drawn from ± a bounded random value ($c; 0 \le c \le 0.5$). Noiseless (algorithmic) advice means that $c = 0$. A graphical representation of the individual decision-making model is presented in Fig 1, below. The figure shows that bias is modeled to have the strongest impact on updating decision-makers with intermediate (uncertain) prior beliefs. Fig 1 also shows that noise makes the updating of beliefs substantially less predictable.

## Agent-based model

To study *how* noise and bias in advice impact decision-makers' behavior and population-level outcomes, we built an agent-based model (ABM). Models of human decision-making have a long history in social and behavioral sciences. For much of this history, progress was limited by the requirement of omnicompetent decision-makers who worked with perfect recall, no bias, and no noise [19]. This research paradigm, known as the *Bayesian rationality approach* (BRA), was upended by the seminal heuristics and biases program, which found over and over again that human behavior violated BRA assumptions [20,21]. For example, the bias of 'anchoring' makes individuals' judgments and decisions highly sensitive to an initial value; a bias found to be prevalent in policing contexts [22], Scholars have also argued that that systematically biased noise offers an alternative explanation for the findings of heuristics and biases experiments [23].

Agent-based models effectively link individual decision-making to the population-level behavioral outcomes [24]. Agent-based models can also incorporate rich details that persuasively resemble the complexity of a real-world environment [19]. Our use of agent-based modeling builds on the insights of Choi and Park (2021), who advocate for ABMs as powerful tools for theory development in public administration, particularly for modeling complex systems with emergent behavioral outcomes [25]. The functionality of ABMs allowed us to model stochastic spatial interactions that are, are characteristic of

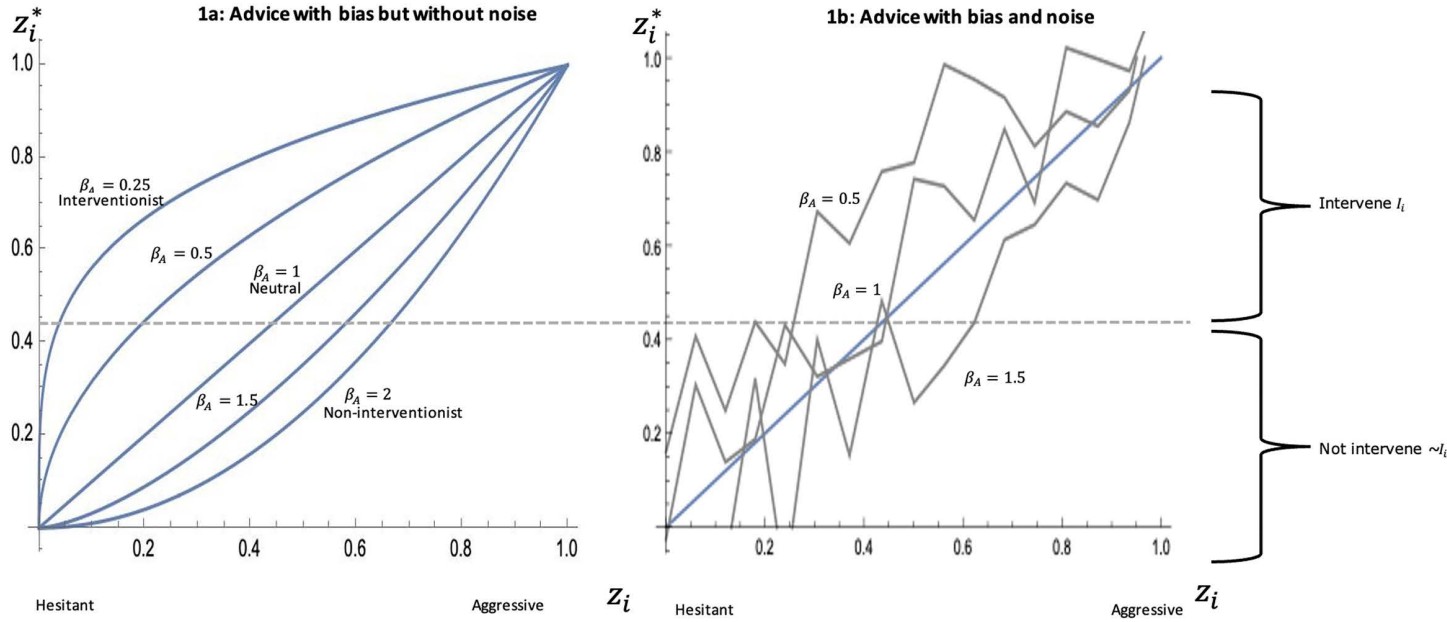

**Fig 1. Transformation of the decision-makers' prior into posterior beliefs. (A)** The value on the x-axis is the decision makers' prior belief $z_i$ which is transformed into the posterior belief $z_i^*$ on the y-axis. This transformation is through the biased (or neutral) advice $\beta_A$, received by the decision-maker. In Fig 1A four different values of bias of advice are plotted ($\beta_A = 0.25, 0.5, 1, 1.5, 2$). The horizontal middle line indicates the cutoff point for the decision-maker to intervene $I_i$ or not intervene $\sim I_i$. **(B)** A random shock $\varepsilon_i$ with ($c = -0.25, 0.25$) is introduced to advice bias with values $\beta_A = 0.5, 1, 1.5$. The gray lines are one representation of random noise; they are not deterministic. The linear function (blue line) in Fig 1B presents, for comparability, noiseless, neutral advice.

patrol-style and other repeated-encounter environments. In our model, $N=4$ decision-makers interact with a population of $N=1,000$ subjects. In each model run, the decision-makers must determine which of the subjects they encounter (in 6,000 'time steps') are 'bad'. The agent-based model aggregates the behavior of the four decision-makers encountering multiple subjects while randomly patrolling a two-dimensional environment and receiving advice that varies in both bias $\beta_A$ and noise $\varepsilon_A$, with algorithmic advice assumed to be noiseless. A graphical representation of our model is shown in Fig 2.

Agent-based models, unlike other modeling methods, can incorporate populations of decision-makers and subjects with heterogeneous characteristics [26]. We make use of this capability in our model by introducing a heterogeneous population of subjects who are either 'bad' or 'not bad' (good). We also carefully devised different distributions of decision-makers' prior beliefs $z_{id}$. The distribution in Equation (1) modeled the baseline of a random uniform draw, which we subsequently compared with biased normal distributions of prior beliefs. One biased normal distribution represented a population of decision-makers with *hesitant* prior beliefs ($\mu=0.25$). Another biased normal distribution represented a population of decision-makers with *aggressive* prior beliefs ($\mu=0.75$). We also modeled a *polarized* distribution of biased priors of decision-makers with either $\mu_1=0.25$ or $\mu_2=0.75$ in the simulations (additional model details in Methods). Given the tendency for polarization in society and representative government [27,28], models of decision-making should also consider the impact of polarized prior beliefs held in populations of decision-makers [29].

We varied advice bias $\beta_A$ over nine values across the ABM simulations between extreme *interventionist* ($\beta_A=0$) and strongly *non-interventionist* ($\beta_A=2$). We also varied advice noise $c$ (0, 0.25, 0.5) across the ABM simulations. We repeated each run 50 times (with 6,000 time steps in each run) to obtain sufficient variation in decision-makers' aggregate behavioral outcomes under similar conditions. In each model run, 10% of the model population ($N=1,000$) is deemed as *not*

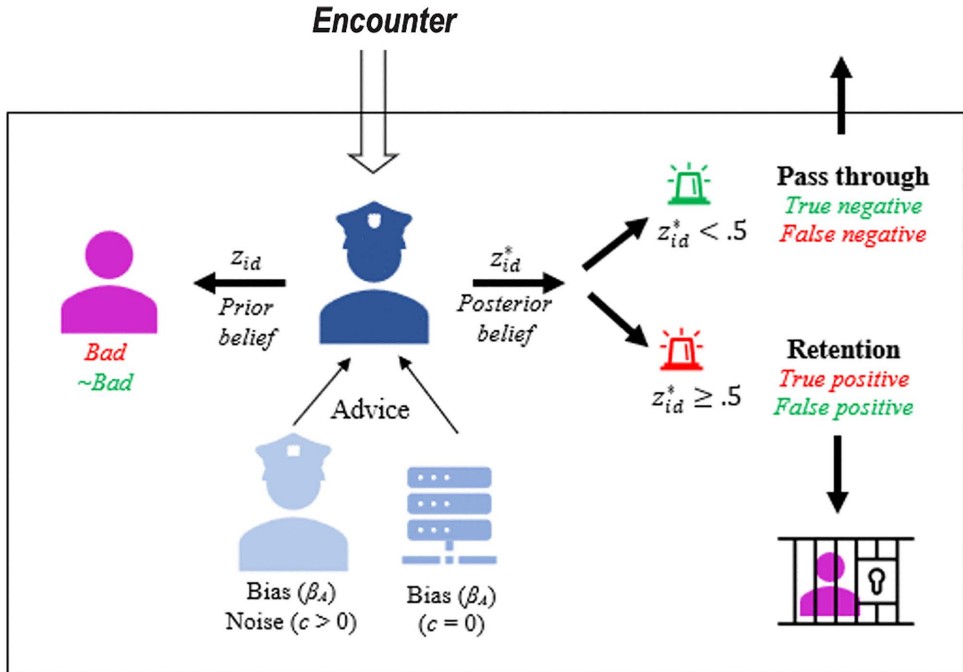

**Fig 2. Modelling encounters between decision-makers and subjects.** At the beginning of every model run both subjects and decision-makers wander randomly (in all directions) around the model environment. There are $N=1,000$ subjects and $N=4$ decision-makers at model start. A random 10% subset of the subjects are determined to be~bad at the model start while the rest are bad. If a subject and a decision-maker *encounter* each other and interact the decision-maker decides whether to put the subject into retention based on a combination of their prior belief combined with the advice they receive which may or may not be biased ($\beta_A$) or noisy ($\varepsilon_A$). A verification process then reveals whether the decision maker made an accurate (true positive/negative) or inaccurate decision (false positive/negative).

*bad*, the rest as *bad*, a percentage that does not influence model dynamics but serves as a benchmark for evaluating behavioral outcomes. Importantly, varying this bad or not bad percentage affects only the distribution of false and true positives in the simulation results. The percentage of bad subjects in the population does not alter the combined retention total (i.e., the sum of true and false positives; see S1 File). We chose this modelling approach in which the not bad or bad labels are stable designations within each model run in order to isolate and track the impact of bias and noise on behavioral outcomes without introducing additionally complexity within our subject population. The simplest behavioral outcome is the median number of subjects who faced intervention from decision-makers (put in retention) after 6,000 time steps. Those subjects include both true positives (*TP*) as well as false positives (*FP*). Subjects correctly missed are true negatives (*TN*), while those falsely missed are false negatives *(FN)*. Thus, our simulation approach provided us with the requisite data to compare behavioral outcomes in our model across a broad range of conditions defined by distributions of prior beliefs, advice bias, and advice noise.

## Results

### Effects of advice bias and noise when decision-makers' priors are randomly distributed

We conducted a series of simulations to analyze the impact of advice bias and noise on the behavior of four decision-makers. The first simulations exclusively focused on the effects of advice, holding constant a uniform distribution of decision-makers' prior beliefs. Advice bias $\beta_A$ was varied across nine levels, ranging from 0 (strictly interventionist) to 2 (strongly non-interventionist), with 1 representing neutrality. We varied *advice noise* by running these simulations for three different values of parameter $c$ of the random noise function, reflecting no noise ($c=0$), moderate noise ($c=0.25$), and high noise ($c=0.5$). The runs with no noise aimed to simulate the effects of algorithmic advice. Runs with higher levels of noise aimed to simulate human advice with increasing levels of noise. We did 50 simulation runs of each of the 27 parameter combinations to obtain enough variation in the simulated population-level behavior of decision-makers. Results are presented in Fig 3.

Fig 3 shows the effect of different values of advice bias (horizontal axis) on three behavioral outcomes: number of true positives, false positives, and false negatives, respectively. For each value of bias, three boxplots present the median and IQR of the 50 simulations for noiseless advice ($c=0$) and the two levels of noisy advice ($c=0.25$; 0.5). The results at ($\beta_A=1$) represent the behavioral outcomes under neutral advice. Because the prior beliefs of decision-makers are uniformly distributed, we observe above and under ($\beta_A=1$) results which diverge from mere chance. Fig 3 leads to three key observations. The first observation is that advice bias strongly affects behavioral outcomes. The pattern is slightly non-linear. The pattern of true positives is similar to the pattern of false positives, and inverse to the pattern of false negatives.

The second observation follows from comparing the medians in the boxplots. Noise slightly pushes the outcomes towards those under neutral advice. We also observe interesting interaction effects between bias and noise on behavioral outcomes. Adding noise to advice moderates (softens) the impact of advice bias on behavioral outcomes, especially at the extreme ends of the advice bias spectrum. At those ends, the impact of bias on algorithmic advice is more pronounced. This has implications for specific behavioral outcomes. Under advice with an interventionist bias, noise lowers both the number of true and false positives, while for non-interventionist advice, these numbers increase (see Fig 3A, 3B). For false negatives, the impact is reversed.

A third observation from Fig 3 is that the size of the IQR of the boxplots does not vary across different values of $c$. This has an important implication. The variability of population-level behavioral outcomes of decision-makers does not increase when advice is noisier. Stated differently, our simulations challenge the assumption that noiseless (algorithmic) advice inherently reduces variability in population-level decision-making.

### Effects of algorithmic advice under biased priors of decision-makers

In a second series of simulations, we explored how advice bias and noise would affect the behavior of a population of decision-makers with biased prior beliefs. We drew prior beliefs from a normal distribution $z_i \sim \mathcal{N}(\mu, \sigma^2)$. Varying the

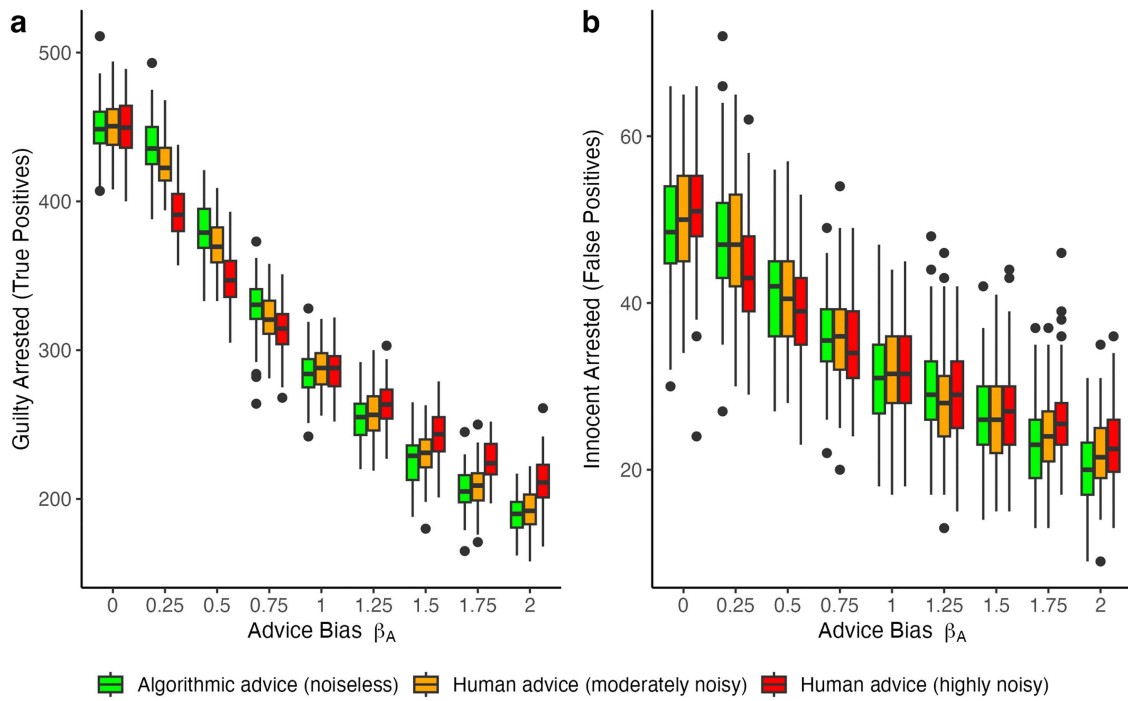

**Fig 3. True and false positive results with varying levels of noise and unbiased advice. (A)** Count of *true positives* across nine levels of advice bias $\beta_A$. The colored boxes represent the median (line) and the interquartile range (IQR) of 50 simulation runs, holding constant the parameter combinations ($\beta_A$, *c*). There are 1000 subjects in each model run and 10% are not bad, while the rest are bad. The lines above and below the boxes represent ± 1.5 * IQR. The points above or below the IQR are considered outliers. Colors represent different levels of advice noise (algorithmic advice: *c*=0; human advice *c*=0.25 and *c*=0.5 respectively). **(B)** Same plot except for the behavioral outcome which is the count of *false positives* across nine levels of advice bias $\beta_A$. Other than the counts as behavioral outcomes (and therefore the range), the three plots are constructed using the same approach.

value of *μ* reflects that, at the population level, decision-makers' priors can be systematically biased. Such bias could be towards the *hesitant* prior belief that subjects are 'not bad' (*μ*=0.25), or the more *aggressive* prior belief that subjects are 'bad' (*μ*=0.75). As a baseline, we drew from a distribution of prior beliefs with *uncertainty* as the expected value (*μ*=0.5). This allows for comparing this second series of simulations with the first series. In addition, we drew from a bimodal distribution $z_i \sim 0.5 \cdot \mathcal{N}(\mu_1, \sigma^2) + 0.5 \cdot \mathcal{N}(\mu_2, \sigma^2)$ to reflect a population of decision-makers with *polarized* prior beliefs, where $\mu_1 = 0.25$ and $\mu_2 = 0.75$ (see S1 File for further details).

We first performed a series of simulations with advice noise *c*=0, representing noiseless (algorithmic) advice. The results of these simulations are presented in Fig 4. The boxplots now represent the four different distributions of decision-makers' prior beliefs. Above and under the results at ($\beta_A$=1) we observe outcomes that diverge from mere chance. We did not plot the pattern of false negatives, because it is the reverse of the pattern of true positives. Fig 4 leads to two new observations. The first new observation is that patterns of behavioral outcomes rapidly diverge for populations of police officers with biased priors. (The pattern of the baseline population of uncertain decision-makers (*μ*=0.5) is similar to those in Fig 3). Unsurprisingly, the behavior of aggressive decision-makers is mostly affected by higher levels of opposite (non-interventionist) algorithmic advice ($\beta_A$>1). Similarly, the behavior of hesitant decision-makers is mostly affected by higher levels of opposite (interventionist) algorithmic advice (0<$\beta_A$<1). Thus, the simulations show that decision-makers' behavior in biased populations will only change under relatively extreme, opposite algorithmic advice.

Fig 4 reveals a second, remarkable observation. The behavior of decision-makers in a population with polarized prior beliefs follows a unique pattern. For this bimodal distribution, we expected a pattern that would roughly resemble

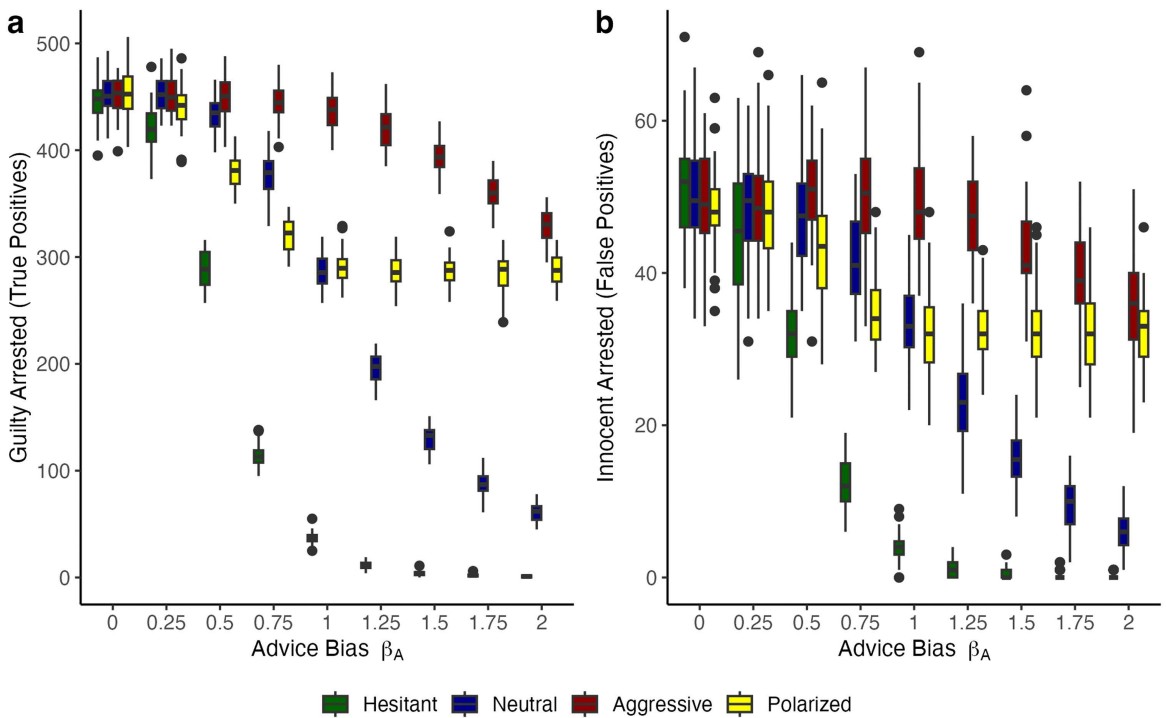

**Fig 4. True and false positive results with varying levels of bias and noiseless advice. (A)** Count of *true positives* across nine levels of advice bias $\beta_A$. The colored boxes represent the median (line) and the interquartile range (IQR) of 50 simulation runs, holding constant the parameter combinations $(\beta_A, c)$. There are 1000 subjects in each model run and 10% are not bad, while the rest are bad. The lines above and below the boxes represent ± 1.5 * IQR. The points above or below the IQR are considered outliers. Colors represent different levels of population bias in decision-makers' prior beliefs (hesitant, neutral, aggressive and polarized). There is no advice noise. **(B)** Same plot except for the behavioral outcome which is the count of *false positives* across nine levels of advice bias $\beta_A$. Other than the counts as behavioral outcomes (and therefore the range), the two plots are constructed using the same approach.

that of the uncertain population. However, the pattern of the bimodal population is unique. On the interventionist side of advice bias ($0 < \beta_A < 1$), behavior is less affected by bias than that of the population of uncertain decision-makers. On the non-interventionist side of advice bias ($\beta_A \geq 1$), the algorithmic advice bias does not affect the behavior of the polarized population. The numbers remain on a plateau, which is comparable to that of full uncertainty. We further explore this phenomenon of *advice asymmetry in polarized populations* in the discussion section and provide additional analyses in the S1 File.

### Effects of human advice under biased priors of decision-makers

We further performed a series of simulations with advice noise $c > 0$, representing noisy (ambiguous human) advice. Because results are more pronounced for the simulations under a high level of noise ($c = 0.5$), we present these in Fig 5. The results for a moderate level of noise ($c = 0.25$) can be checked in the Supplementary Materials. Fig 5 reveals a different pattern in behavioral outcomes compared with Fig 4. The effect of advice bias is much less pronounced under noisy (ambiguous human) advice. The introduction of (human) advice noise in biased populations of decision-makers generally pushes the patterns of behavioral outcomes in the biased populations (hesitant or aggressive) towards those of the uncertain population. The behavioral patterns resemble those in Fig 3 (priors under a uniform distribution), albeit the curves lean much more toward the uncertain baseline. Thus, adding advice noise turns the population-level behavior of biased decision-makers towards that of a relatively uncertain population of decision-makers with random priors.

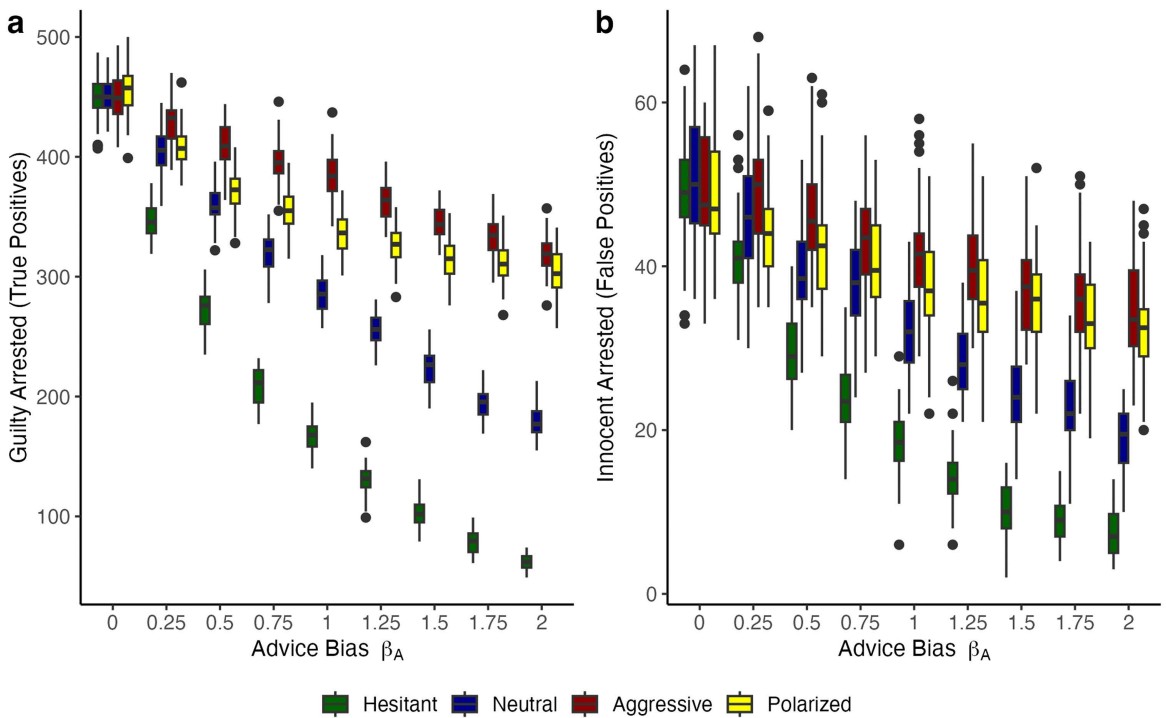

**Fig 5. True and false positive results with varying levels of bias and highly noisy advice. (A)** Count of true positives across nine levels of advice bias $\beta_A$. The colored boxes represent the median (line) and the interquartile range (IQR) of 50 simulation runs, holding constant the parameter combinations ($\beta_A$, $c$). These plots have highly noisy (human) advice for $c$=0.5. There are 1000 subjects in each model run and 10% are not bad, while the rest are bad. The lines above and below the boxes represent ±1.5 * IQR. The points above or below the IQR are considered outliers. Colors represent different levels of population bias in decision-makers' prior beliefs (hesitant, neutral, aggressive and polarized). **(B)** Same plot except for the behavioral outcome which is the count of *false positives* across nine levels of advice bias $\beta_A$ with highly noisy (human) advice $c$=0.5. Other than the counts as behavioral outcomes (and therefore the range), the two plots are constructed using the same approach.

The observations above also hold for the polarized population. Yet, the asymmetry in behavioral outcomes for this population in Fig 4 (algorithmic advice) is still clearly discernible in Fig 5. We observe the plateau effect for the polarized population of decision-makers around $\beta_A \geq 1.5$, again converging to numbers associated with full uncertainty.

## Discussion

### Bias in advice substantially affects behavioral outcomes

In our agent-based model, decision-makers encounter subjects with sets of characteristics in a two-dimensional environment and decide whether to refer a subject for further screening. This representation can be envisioned as police officers on patrol in a neighborhood who seek advice in their decision to apprehend a (suspected) person. This model replicates the results of extant studies, namely that the level of bias in advice can affect population-level behavioral outcomes [30–32]. For example, He & Pan (2024) report that selfish or fair advice leads participants in experimental dictator games to behave significantly more selfishly or fairly. Our simulations reproduce such effects consistently for different distributions of decision-makers' prior beliefs and for different levels of noise in the advice.

### Environmental noise produces random variation in behavioral outcomes

Kahneman et al. (2021) define noise as "undesirable variability in judgments of the same problem." Their general understanding is that noise is bad: it will turn decision outcomes unpredictable or systematically flawed [2,3]. In our agent-based

model, three different sources of noise in advice moderated the relationship between decision-makers' updating of prior beliefs and their population-level behavior. We introduced random variation in: (1) the prior beliefs of decision-makers, (2) the advice given to them, and (3) their behavior in the environment of subjects. Each of these can moderate how prior beliefs are updated and influence population-level behavior. Due to the potential complexity of these effects, we ran extensive simulations systematically varying each source.

From the noise framework introduced by Kahneman et al. (2021), it follows that individual-level sources of noise would create variability in population behavior. Conversely, system-level noise could be decomposed into individual-level sources of noise [2]. However, and surprisingly, in our model simulations, noise in population-level behavior emerged exclusively from *noise in the environment*, generated by the random walks of decision-makers and subjects. This random variation can be interpreted as occasion noise in the environment, a system-level factor that shapes aggregate behavior. Imagine, for example, a sudden downpour that leads a police officer on patrol to duck into a coffee shop. This shelter prevents encounters between officers and potential suspects unpredictably (also when suspects coincidentally gather in the same coffee shop). Empirical evidence indeed shows that warmer temperatures are associated with more arrests [33].

To further validate the attribution of random variation in population-level behavioral outcomes to environmental noise, we built a benchmark model that excludes such environmental noise (implemented in RStudio; see S1 File). The benchmark model can be envisioned as subjects queuing in line for decision-makers: e.g., passengers waiting in line for immigration officers at the airport. The benchmark model shows less random variation in population-level outcomes (see S1 File). Clearly, controlled work environments reduce random variation in population-level behavioral outcomes.

## Algorithmic advice does not reduce variability in behavioral outcomes

Our simulations revealed that larger levels of random noise in *prior beliefs* or *advice* do not produce a greater variability in population-level behavioral outcomes (see S1 File). The explanation is that intra-decision-maker noise canceled out between different encounters/decisions. System-level random variation can cancel out, and scholars argue it often will, which may obfuscate the individual-level sources of noise [3,23,34]. The repeated interactions of decision-makers with subjects buffered the impact of individual-level noise on population-level random variation in behavior. Thus, algorithmic advice does not necessarily reduce the variability of behavioral outcomes at the population level in our model. This challenges the assumption that system-level variation can be easily traced back to individual-level noise and provides further theoretical justification for authors who warn that the absence of system-level noise does not necessarily indicate that individual-level noise is absent [3]. These simulation results refute the claim that system-level noise can be easily decomposed into sub-level and/ or individual-level sources of noise [2,35,36].

## Noise in advice makes decision-makers revert to their prior beliefs

While noise in advice did not increase variability in population-level outcomes, it nevertheless systematically impacted behavioral outcomes. When we added noise to advice in our simulations, decision-makers behaved more like those we observed under *neutral advice*. Thus, the simulations showed that increased levels of noise in advice leads decision-makers to systematically revert to their prior beliefs. It is important to note that in our simulations, this reversion to prior beliefs is an emergent property of decision-making under noisy advice. We observe this emergent behavior in all model specifications, including those where decision-makers have biased prior beliefs. Decision-makers' reverting to prior beliefs under noisy advice could be interpreted as a manifestation of *confirmation bias* (which is, however, not part of the advice updating function in the model) [37]. This suggests that confirmation bias can be understood as an emergent property of the model rather than a programmed assumption. This emergent property underscores that noiseless (algorithmic) advice has a stronger effect on the adjustment of prior beliefs than human advice. Additionally, this effect is more pronounced near the endpoints [0, 2] of advice bias $\beta_A$. This is due to the random shocks of noise pushing the results towards a

"ceiling" and therefore away from the endpoint. This is known as a *ceiling effect* and is realistic in decision-making, since one cannot be more than 100 percent certain when choosing between options.

Reversion to prior beliefs provides an explanation for the results of empirical studies into differences between judicial decisions. One study reported asylum applicants have an 88 percent chance of prevailing in front of one judge and a 5 percent chance of prevailing in front of a different judge in the same city [38]. This finding could be interpreted as (strict or lenient) judges reverting to their prior beliefs by denying advice (e.g., policy guidelines, precedent) from an overwhelming majority of asylum applications. Another study reported that algorithms applied to previous bail decisions were estimated to reduce jailing rates by over 40 percent without increasing crime rates [11]. If advised by those algorithms, relatively strict judges could have been aided to move off their prior beliefs and, therefore, safely release more individuals.

### Noise in advice may sometimes be preferable over algorithmic advice

Our simulations showed that (strongly) biased algorithmic advice has the largest effect on population-level behavioral outcomes. At the ends of the bias scale (non-interventionist or interventionist), we observe an increasing impact on false positives or false negatives. These effects of algorithmic bias on erroneous decision-making – at relatively extreme levels of bias – are moderated by noise in advice. When the advice is strongly biased, decision errors may be reduced when decision-makers use noisy (human) advice rather than noiseless (algorithmic) advice. An empirical study on the use of algorithmic technology in the criminal justice system reported that such algorithms have been "less helpful" and even "harmful" [16]. Our model simulations attribute those negative effects more specifically to situations in which algorithmic bias is strong. The conditional error reduction through noise in advice could also provide an explanation for the results of empirical research showing that the integration of human decision-making with algorithmic decision-making does not (always) lead to synergistic results [39].

Our study extends emerging research which highlights the possible benefits of noise in (collective) decision-making contexts. This includes empirical findings of improved collaborative performance and consensus building [8,40] and theoretical research which argues that noise is an important feature of human cognition enabling humans to sample diverse hypotheses in response to an uncertain world [41]. In addition, our model simulations show that noise in advice reduces decision errors when that advice is strongly biased.

### Polarized decision-makers asymmetrically react to biases in advice

A surprising result of the model simulations was the unique, asymmetrical pattern of behavioral outcomes exhibited by the polarized decision-makers. The polarized population of decision-makers was only sensitive to interventionist advice. Under increasingly more interventionist advice bias, the polarized population rapidly exhibited a behavioral pattern similar to the decision-makers with more aggressive prior beliefs. Under increasingly non-interventionist advice bias, the polarized population showed a pattern of behavior reflecting full uncertainty. Full uncertainty refers to a population of decision-makers with an expected prior belief of full uncertainty ($E(z_i) = 0.5$) and neutral advice ($\beta_A = 1$).

This asymmetry is not due to an artifact in the aggregation: inspection of the simulated individual behaviors of decision-makers reveals a markedly different response to non-interventionist than to interventionist advice. When advice is non-interventionist ($\beta_A > 1$), the polarized decision-makers no longer update their behaviors. Mathematically, the updating function exhibits different curvature behaviors: it is relatively more concave in one region ($\beta_A < 1$) and convex ($\beta_A > 1$) in another (see S1 File). The *aggressive* decision-makers in the polarized distribution predominantly pull the overall behavioral outcomes in their prior direction, while the *hesitant* decision-makers do not (see Supplementary Material). By contrast, under separate distributions (biased prior beliefs) of hesitant and aggressive priors, advice impacts on behavioral outcomes in response to both interventionist and non-interventionist advice. Within the polarized distribution of decision-makers, the reaction to interventionist advice more strongly influences their population-level behavior, drowning out the impact of non-interventionist advice.

The asymmetry in emergent behavioral patterns echoes findings from the motivated reasoning literature in political science. One study revealed that, in response to COVID-19 public health advice in the United States, polarized Republicans stopped updating their behavior. This was after an initial period of deep nationwide concern [42]. Under Democratic Party governance, they quickly reverted to the Republican partisan prior belief of skepticism and mistrust in government institutions: not only public health institutions but also scientific studies offering health advice [43,44]. Scholars later found Republicans had worse health outcomes at the population-level, which (in part) pulled the United States towards a poor overall COVID-19 response [45,46]. Another study, in immigration politics, showed experimental evidence that conservatives react more strongly to fear-based or crime-related messaging [47], a strategy that is common in U.S. political advertising [48]. This type of fear framing functions as a biased advice to citizens' decision-making, amplifying restrictive preferences of conservatives specifically. Thus, fear-framing helped producing immigration policies that are considerably more adversarial than public opinion would suggest [49,50].

## Limitations

Agent-based modeling offers many benefits, such as the ability to capture emergent phenomena, increased flexibility, and the ability to theoretically explore the effects of complex conditions [51]. Our use of agent-based modeling does not aim to test empirical validity beyond theoretical exploration [52,53]. That said, model assumptions and parameters are empirically calibrated. For example, our decision-maker-to-subject ratio is based on FBI-collected police officer/population ratios [54]. A limitation of agent-based modeling is the simplification of human behavior into simple decision rules, rules that do not necessarily incorporate realistic friction in decision-making (including cognitive biases such as anchoring and availability) [19].

We assumed that algorithmic advice is noiseless [2,3]. This assumption is debated. While algorithms eliminate noise in the prediction of future cases, a complex algorithmic model could capture past noise in its training data [55]. Moreover, noise in algorithmic advice can arise from the interaction between the decision-maker and the technology. For example, a simple device generating "red" versus "green" advice assumes that the decision-makers are not colorblind. Nevertheless, we contend that it is still sound to assume that algorithmic advice exhibits less noise than human advice so that our results remain valid

## Recommendations for future research

Future research can take both empirical and theoretical directions. The empirical direction would be to develop a series of experiments that test the hypotheses generated by the model. The design of these experiments can be specially tailored to further tease out the conditions under which humans and algorithms produce synergetic effects [39]. Such experiments could systematically vary noise and bias in advice to identify their independent and joint effects. Ideally, those experiments would be embedded in real-life work settings of decision-makers, such as police officers, immigration officers, or medical specialists. While our agent-based model is about behavior and not about prediction correctness, an empirical study could consider the accuracy of decision-making. Kahneman et al. (2021) explain that system noise can be broken into two parts: level noise and pattern noise. However, they propose evaluative methods such as *noise audits* for empirical investigations of noise in organizations, suggesting noise decomposition is more ambiguous in the real world. This raises the possibility that noise decomposition may not only be ambiguous in applied settings, but, in some cases, effectively unobservable. Future research should explore whether and how population-level variation can be meaningfully traced back to distinct sources of noise in real-world contexts.

Our theoretical model departs from the assumption that priors of decision-makers are drawn from an (un)biased random distribution of all possible priors. A theoretical direction for future research would be to extend this simple model to incorporate learning decision-makers (boundedly rational), learning from subjects, and advice. Additionally, future models could incorporate self-learning algorithmic advice to reflect the non-determinism of contemporary AI systems. Recent

research has challenged the concept of *noiseless* algorithms in the context of modern artificial intelligence and machine learning systems [6,7]. These systems are *non-deterministic* and can be *self-learning*, increasing (potentially unwanted) variability (noise) in model predictions [6,56]. A self-learning decision rule would then simulate the impact of machine learning in a real-world artificial intelligence system advising human decision-makers. Another possible extension is to introduce strategic behaviors of decision-makers and subjects. For example, citizens guilty of a crime take measures to avoid a police encounter [57]. Police officers strategically patrol in specific neighborhoods. Rather than being labelled as either good or bad, subjects could have distributions of "cognitive" traits that determine a potential proclivity for criminal activity, which decision-makers attempt to learn. Future model extensions could include costly actions. The police may operate in a locale where the populace is concerned about crime or the consequences of crime enforcement [58]. We could model, for example, a *tough-on-crime* policy in which the police receive an additional penalty (cost) for not arresting a guilty person (false negative) or a *police reform* policy in which the police receive an additional penalty for arresting an innocent person (false positive). Additionally, scholars have proposed that noise distributions are more complex than typically modeled [41]. Future modeling and empirical research can incorporate these findings.

## Implications for practice

The model simulations presented in this paper have clear implications for practice. Biased algorithmic advice exerts a stronger influence on behavioral outcomes than equally biased human advice. This replicates practical experience. For example, in Miami, a Colombian asylum applicant was almost 17 times more likely to be allowed to stay in the country if they faced the most lenient versus the strictest judge in the building [35]. In contexts marked by high human variability, algorithmic support may offer gains in consistency and fairness. Algorithms eliminate noise (in advice, in our model), which can increase the consistency of decision-making (for example, judicial rulings). They can also more effectively moderate the prior beliefs of decision-makers. Practitioners should consider (and carefully test) algorithmic solutions under these conditions.

The noise-eliminating (or reducing) nature of algorithms, however, presents a potentially dangerous combination when advice is highly biased. This combination produced the strongest effect on behavioral outcomes in our model. Biased algorithmic advice can, in some cases, lead to worse outcomes than equally biased human advice. Empirical studies reveal that algorithmic advice can lead to harmful outcomes when highly biased, especially in predictive policing [9,59–61]. This harm has implications for the standard of disparate impact, a legal doctrine requiring practices with disproportionately adverse effects on certain groups to be halted [3]. This standard is used by U.S. discrimination law and proposed by Sunstein (2022) as a means of assessing whether an algorithmic solution should be implemented. Our results suggest that algorithmic advice can produce disparate impacts even in situations where there is already human bias in the system, when the training data is highly biased. Clearly, professionals must extensively test any proposed algorithmic system and continue monitoring post-implementation. However, when sufficient testing is not feasible [62], we recommend using noisier human advice, which can surprisingly reduce the impact of bias.

An additional recommendation for policymakers that follows from our simulations is to be aware that the use of algorithmic advice inevitably pushes to the background the prior beliefs of professional decision-makers. This behavior should not be misinterpreted as technology aversion or acceptance; it emerges from belief updating under structured advice [63]. Our model simulations reveal that it is rather an emergent property of the simple updating of prior beliefs by professionals based on algorithmic advice. A proper balance between professional judgment, collegial advice, and algorithmic support will contribute to synergy, albeit under the specific conditions derived from our model simulations.

Furthermore, our findings raise questions about the feasibility of *noise audits* that seek to attribute population-level variation to individual sources of noise. In our simulations, variability in outcomes was primarily driven by occasion noise in the environment, while deliberately introduced noise in advice did not produce additional variability. This result contrasts with proposals that organizational variability can be decomposed into individual-level noise, as suggested by

Kahneman et al. (2021). Recent applications of noise audits in policing show that such methods can detect systematic divergences across officers [64], and medical decision-making research highlights how different forms of noise (level, pattern, occasion) often interact in ways that are difficult to disentangle [5]. Given the evidence of positive effects from noise [8,40] a more cautious approach is wise. Taken together, these literatures suggest that while noise audits may provide useful organizational insights, their capacity to separate system-level from individual-level variability may be more limited than often assumed. Moreover, one should be careful to interpret the effects of noise on decision-making exclusively in a negative way.

Finally, our model offers guidance for dealing with a polarized populace, a common situation in many developed countries [65]. Our findings underscore that polarization can impact advice and decision-systems, regardless of their content. In our model, biased advice to polarized decision-makers led to asymmetrical results. At the population-level, the effect of *non-interventionist* advice was imperceptible. The *aggressive* decision-makers in the population simply drowned out the impact of the *hesitant* decision-makers, even when the advice was *non-interventionist*. This pattern suggests that well-intended interventions may fail to produce balanced behavioral outcomes in polarized populations, unless variation in responsiveness is explicitly considered. In Europe (as in the United States), political polarization was associated with higher excess mortality during the COVID-19 pandemic [42,47,66]. It is not just the pandemic; political polarization is generally associated with worse health outcomes [67].

We observe a comparable pattern in immigration politics. A majority of Americans have consistently rejected the idea of deporting all undocumented immigrants [49,68]. Experimental evidence shows that conservative respondents are significantly more likely to endorse restrictive immigration attitudes when exposed to fear- or crime-based biased messaging [47]. This asymmetric responsiveness to threat cues helps explain the United States' increasingly hardline immigration stance and policies with only one-third of the population backing such a policy [50].

## Materials and methods

### Individual-level model of advice

The individual-level model that informs our agent-based model is inspired by a canonical formal model, in which a politician receives advice from a (biased) advisor [17,18]. Calvert (1985) shows that biased political advice influences a politician to change prior beliefs about the best choice among two options. In our individual-level model, a decision-maker $i$ must make a choice regarding a particular subject ($s_d$). Subjects are in either one of two states: 'bad', where $s_d$ {$B$}, or 'not bad', where $s_d \in$ {$\sim B$}. A bad state requires intervention, and the decision-maker's task is to decide whether to intervene or not.

The decision set is binary: the decision-maker, in their interaction with the subject, has the choice either to intervene ($I_i$) or not to intervene ($\sim I_i$). Decision-makers are assumed to base this choice on two sources of information. The first source of information is a set of observable characteristics $x$ {$x_{1d}, \ldots, x_{kd}$} of subject ($s_d$) that informs the decision-maker whether (or to what extent) the subject is bad. Based on these observable characteristics, the decision-maker $i$ formulates a prior belief ($z_{id}$; $0 \leq z_{id} \leq 1$) about subject $s_d$. This prior belief is an expression of the decision-maker's uncertainty about the state of the subject. The prior belief $z_{id}$ is the probability that decision-maker $i$ attaches to subject $s_d$ that they are bad, that is, $p_i(s_d \in \{B\})$, or not bad, $p_i(s_d \in \{\sim B\})$. In the decision-makers prior belief, uncertainty about the state of the subject factors into the decision-making process. When the decision-maker is fully uncertain about the state of the subject, the decision-maker will metaphorically flip a coin to decide whether to intervene or not, then $z_{id} = 0.5$.

When the decision-maker is more certain about the state of the subject, the prior belief will diverge from 0.5. The more certain a decision-maker is about the subject being bad ($s_d \in \{B\}$), the higher the prior belief ($0.5 < z_{id} \leq 1$). Decision-makers with prior beliefs $z_{id}$ larger than 0.5 are *aggressive*. Decision-makers with high prior beliefs would initially choose to intervene ($I_{id}$) in subject $d$. By contrast, the more certain a decision-maker is about the subject being not bad ($s_d \in \{\sim B\}$),

the lower the prior belief ($0 \leq z_{id} < 0.5$). Decision-makers with low prior beliefs $z_{id}$ are *hesitant* and will choose not to inter-vene ($\sim I_{id}$) in subject *d*.

In the model, decision-makers update their prior beliefs once, based on advice (*A*) by either a human or an algo-rithm. After advice is given, and their prior beliefs are updated, decision-makers decide whether to intervene or not. In our model, advice is the expression of an external opinion towards intervention or no intervention. We assume that advice is bilateral and that the decision-maker has autonomy over their own choices. We assume one advisor in each model run (eliminates *level* noise) with a stable level of bias (advice bias). We assume algorithms exhibit no noise and will make the same prediction based on similar subject characteristics every time $s_d$ is encountered. Human advice is assumed to be both noisy and biased. We model the *pattern* and *occasional* noise inherent in human advice (see noise parameter *c* below).

Advice has two main characteristics. The first characteristic of advice is *bias* ($\beta_A$), where $\beta_A \geq 0$. The advice can be biased towards a more *interventionist* choice ($0 \leq \beta_A < 1$) or towards a more *non-interventionist* choice ($\beta_A > 1$). The advice can also be *neutral* ($\beta_A = 1$). Advice alters the decision-makers prior belief $z_{id}$ into a posterior belief $z_{id}^*$. This transformation is defined by a simple advice update function $U(z_{id})$.

$$z_{id}^* \;=\; U(z_{id}) \;=\; z_{id}^{\beta_A} \tag{2}$$

The decision-maker is assumed to make a choice based on the posterior belief. If the posterior belief is larger than or equal to 0.5, the decision-maker will choose to intervene: $p(I_{id}) = 1$. In that case, the expected utility of intervening after the advice is larger than the expected utility of non-intervening after the advice. A posterior belief smaller than 0.5 results in the choice not to intervene: $p(\sim I_{id}) = 1$.

$$p(I_{id}) = 1 \quad \text{if } z_{id}^* \geq 0.5; \text{ else } P(\sim I_{id}) = 0 \tag{3}$$

$$p(\sim I_{id}) = 1 \text{ if } z_{id}^* < 0.5; \text{ else } P(I_{id}) \quad = 0$$

Assuming the decision-maker has some autonomy over choice implies that a "command" can be interpreted as an extreme form of advice. In the case of a command to intervene, the decision-maker updates the prior belief to the absolute posterior belief ($z_i^* = 1$) irrespective of the value of prior belief $z_i$. This only occurs under the extreme value of advice bias $\beta_A = 0$. A decision-maker who is "commanded" to intervene updates the prior belief into a posterior belief $z_{id}^* = 1$ for all $z_{id}$.

Fig 1A illustrates that biased advice has a greater impact on intermediate values (more uncertain) of prior belief than on values at the end of the scale. This characteristic of the advice update function follows the intuition that uncertain decision-makers are more sensitive to biased advice than more certain decision-makers.

## Noisy advice and unpredictable posteriors

Advice can be more or less ambiguous, depending on the source of the advice. Ambiguous advice is defined as advice that comes with noise. Algorithms, we assume, produce advice without noise. Although their advice can be biased, it is unambiguous. Human advice can be noisy, depending on personal characteristics and contextual circumstances. Unpredictable variation will enter decision-makers' updating from their prior belief into a posterior belief after noisy advice. To model noisy advice, we extended the advice update function in equation (2) (cf. Callander, 2011). In this extended model, the decision maker still receives neutral or biased advice, depending on the value $\beta_A$. However, the advice now comes with an additional stochastic shock, $\varepsilon_A$, which is a random value between bounds set at *c* and $-c$, where $0 \leq c \leq 0.5$. These bounds restrict the range of the random shock function. Equation (4) presents the modified advice update function.

$$z_{id}^* = U(z_i) = z_{id}^{\beta_A} + \varepsilon_A \sim (-c, c) \text{ where } 0 \leq z_i^* \leq 1 \tag{4}$$

Fig 1 shows the new model graphically, with a linear function for comparability (noiseless, neutral advice). The gray lines in this model are not deterministic and show only one possible representation of the function. Noise in advice makes the decision-making process less predictable. While bias in the advice ($\beta_A$) still has an impact, the added noise makes $z_{id}^*$ less dependent on both the (biased) advice and the prior belief $z_{id}$. The curves in Fig 1B are produced with values for bounds $c = 0.25$. With increasing noise ($c > 0.25$) both the advice bias $\beta_A$ and the prior belief will have less impact on the posterior belief (see S1 File for additional model plots). Without noise ($c = 0$), the advice function in equation (4) is equivalent to equation (2). Given algorithms produce noiseless advice, Fig 1A thus represents the algorithmic advice function. Any value $c > 0$ in the model would represent human advice.

Finally, note that noise has *ceiling effects* at the upper-right and lower-left corners of Fig 1B. Because $0 \leq z_{id}^* \leq 1$, the random shock $\varepsilon_A$ of the advice is truncated for confident decision-makers who receive very interventionist advice. The random shock is also truncated for hesitant decision-makers who receive very non-interventionist advice. This ceiling effect is intuitively plausible: a decision-maker cannot be more aggressive, or more hesitant, than certainty dictates. Hence, noise will lower the posterior belief in aggressive decision-makers who receive (strong) interventionist advice. Noise will also raise the posterior belief in hesitant decision-makers who receive (strong) non-interventionist advice. Thus, noise moderates the decision-making of relatively extreme decision-makers who receive similar, relatively extreme advice.

## Population-level model

The individual-level model lays the foundation of an agent-based model (ABM), which aims to explore the aggregate behavior of decision-makers in a spatial environment. The ABM simulates interactions between an advisor, decision-makers, and subjects. We use the NetLogo agent-based modeling software [69,70]. Model files and parameter settings will be made available to facilitate replication. The decision-makers *wander* in a spatial environment, searching for subjects. The ABM simulates these interactions under various distributions of decision-makers' prior beliefs and for different values of advice bias and noise. The wandering spatial environment simulates – for example – police officers patrolling in the streets looking for suspicious subjects.

The population-level ABM requires the specification of several parameter settings. Given computational restrictions, some parameter refinements are necessary to reduce complexity. There are four decision-makers ($i, j, k, l$) who have the task to intervene in a population $N = 1,000$ subjects. They must select those within this population who are bad ($B = 900$). Thus, the proportion of bad subjects within the population $\pi_B = B/N = .9$ in the simulations.

The ABM simulates the aggregate behavior of multiple decision-makers (a) with distributions of prior beliefs among decision-makers, and (b) receiving advice that varies in both bias and noise. Each run of the model simulates a set of actions of decision-makers in time steps. In a 'wandering' spatial representation, each time step, both decision-makers and subjects take a random walk in two-dimensional space. The internal spatial measure of the NetLogo modeling environment is the "patch". We standardize all of our model runs in an environment where, in every cardinal direction, the world extends 100 patches from the center (patch 0,0). Thus, our environment is 201 x 201 patches (we consider alternative spatial environments in the S1 File).

When a decision-maker encounters a subject, this decision-maker adopts a prior belief $z_{id}$ from the distribution of prior beliefs. The decision-maker subsequently transforms the adopted prior belief into a posterior belief $z_{id}^*$ based on (potentially) biased and noisy advice. The belief update is specified in equation (4). Subsequently, the decision-maker decides whether to intervene or not in the subject, based on the decision rule specified in equation (3).

When a decision-maker decides to intervene in a subject, the subject is put in retention (withdrawn from the population) for 100 time steps. Decision-makers do, however, not learn about the true state of the subject. We repeated each step (a

random walk, adoption of the prior, transformation to a posterior based on advice, the decision to intervene) until 6,000 time steps.

In each simulation run, we hold constant the distribution of decision-makers' priors, as well as the bias and level of noise of advice. We simulated several distributions of prior beliefs. A uniform random distribution reflects the baseline of a random draw of priors of decision-makers, with an expected value $\mu = 0.5$, reflecting full uncertainty. In our population-level model, we compare different distributions that represent the prior beliefs that decision-makers attach to the characteristics of the subjects. Normal distributions reflect assumptions about biases in the group of decision-makers. A normal distribution with $\mu = 0.25$ reflects that the group of decision-makers is generally hesitant, while a normal distribution with $\mu = 0.75$ indicates that the group of decision-makers is generally aggressive. A normal distribution with $\mu = 0.5$ reflects that decision-makers are generally uncertain. We also included a bimodal (polarized) distribution of priors of decision-makers with either $\mu_1 = 0.25$ or $\mu_2 = 0.75$ in the simulations.

We varied advice bias $\beta_A$ across the ABM simulations between 0 and 2 in intervals of 0.25 – with 0 reflecting an extreme interventionist bias and 2 reflecting a severe non-interventionist bias. We also varied advice noise $c$ across the ABM simulations. Variation in noise reflects whether the advice would be algorithmic ($c = 0$) or from human origin ($c = 0.25$; $c = 0.5$) with higher values of $c$ reflecting more (human) ambiguity. We repeated each run 50 times to obtain enough variation in outcomes to inform us about heterogeneity in decision-makers' aggregate behavior under similar conditions.

The model simulations produce several outcomes of decision-makers' aggregate behavior. The simplest outcome is the median number of subjects intervened in after the 6,000 time steps. Those subjects – put in retention – include both true positives (*TP*), that is: subjects correctly identified as bad, as well as false positives (*FP*). The latter category consists of subjects that are erroneously put in retention. Subjects that are erroneously missed are false negatives (FN). Finally, subjects that are correctly missed are true negatives (*TN*). By repeating each model run 50 times, we have enough data available to compare median values of *TP*, *FP*, *FN* and *TN*, obtain variance estimates and inspect behavioral patterns at the individual level.

## Supporting information

**S1 File. Specification, calibration and additional analyses.** In this document we provide additional information about the model specification and calibration. Subsequently, we present several sensitivity analyses that relate to characteristics of the spatial environment in the agent-based model. Finally, we present additional analyses that further probe into some of the results we report in our contribution.
(DOCX)

## Acknowledgments

The author gratefully acknowledges the valuable feedback and insightful discussions received from participants at the IRSPM Conference 2025 in Bologna, particularly those in the panel on *Emerging Technologies and the Transformation of the Public Sector*, as well as from colleagues at the NIG Conference 2025 in Ghent, during the *Algorithms & Digital Government* panel. We also thank two anonymous reviewers. Their thoughtful comments greatly contributed to the refinement of the arguments presented in this paper.

## Author contributions

**Conceptualization:** Spencer Poodiack Parsons, René Torenvlied.

**Formal analysis:** Spencer Poodiack Parsons, René Torenvlied.

**Methodology:** Spencer Poodiack Parsons, René Torenvlied.

**Project administration:** René Torenvlied.

**Visualization:** Spencer Poodiack Parsons.

**Writing – original draft:** Spencer Poodiack Parsons, René Torenvlied.

**Writing – review & editing:** Spencer Poodiack Parsons, René Torenvlied.

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
