## [Decision Letter · Decision Letter 0]

28 Oct 2025

Dear Dr. Poodiack Parsons,

Thank you for submitting your manuscript to PLOS ONE. After careful consideration, we feel that it has merit but does not fully meet PLOS ONE’s publication criteria as it currently stands. Therefore, we invite you to submit a revised version of the manuscript that addresses the points raised during the review process.

We look forward to receiving your revised manuscript.

Kind regards,

Matteo Bodini, Ph.D.

Academic Editor

PLOS ONE

**Journal Requirements: **

4. Please upload a copy of Figure 3c, to which you refer in your text on page 9. If the figure is no longer to be included as part of the submission please remove all reference to it within the text.                          

**Additional Editor Comments:**

After evaluating the manuscript and the reviewers’ reports, I agree that the study is technically sound, clearly written, and offers a valuable theoretical contribution. Both reviewers recommend minor revisions. I agree that the Authors should clarify why the percentage of “bad individuals” does not affect model dynamics and expand on the asymmetry in emergent behavioral patterns with at least one additional example and interpretation. Furthermore, the manuscript would benefit from situating its theoretical model within the broader empirical literature on noise in human decision-making, as suggested by Reviewer #2.

Reviewers' comments:

Reviewer's Responses to Questions

**Comments to the Author**

1. Is the manuscript technically sound, and do the data support the conclusions?

Reviewer #1: Yes

Reviewer #2: Yes

2. Has the statistical analysis been performed appropriately and rigorously?

Reviewer #1: Yes

Reviewer #2: Yes

3. Have the authors made all data underlying the findings in their manuscript fully available?

Reviewer #1: Yes

Reviewer #2: Yes

4. Is the manuscript presented in an intelligible fashion and written in standard English?

Reviewer #1: Yes

Reviewer #2: Yes

Reviewer #1: This work is interesting and well presented. I only have a few concerns and further questions.

1) Around line 180 you write that the percentage of bad individuals does not influence the model dynamics. I would like this statement to be better motivated in the text.

2) I would expect more on the asymmetry in emergent behavioural patterns as in line 383 and below: at least one more example, and a more thourough interpretation.

Reviewer #2: This paper theoretically investigates the interaction between noise (random variability in judgments) and bias (systematic deviation) in human–algorithm collaborative decision-making using an agent-based model. Simulating a virtual environment resembling interactions between police officers and citizens, the authors manipulated advice bias (from interventionist to non-interventionist) and advice noise (noiseless algorithmic vs. noisy human) to examine decision-makers’ behavior. The simulations showed that advice noise did not increase behavioral variability but instead produced an emergent tendency for decision-makers to revert to their prior beliefs, suggesting that human noise can mitigate the negative effects of algorithmic bias. Furthermore, highly biased algorithmic advice sometimes led to more erroneous decisions than equally biased human advice, and advice effects became asymmetric under polarized conditions. Overall, the study provides a theoretical demonstration that while algorithmic advice may enhance consistency, it also carries potential risks when bias is strong, offering valuable insights into human–algorithm decision-making.

This study is based on a solid theoretical framework and does not directly investigate actual human behavior. In this respect, it requires careful consideration as to how far the findings can be generalized to real-world human behavior. Nevertheless, the computer simulations presented in the paper are carefully and rigorously constructed, and the resulting theoretical analyses offer originality and meaningful contributions. Therefore, I believe this manuscript has sufficient merit to be considered for publication in PLOS ONE.

That said, I would like to suggest one important point for revision prior to publication. The concept of “noise” is central to this study, and the paper provides valuable theoretical insights into how noise may influence human judgment and behavior. However, there already exist several influential empirical studies that have examined the effects of noise on human decision-making, which are not currently discussed in the manuscript. To strengthen the theoretical implications and enhance the broader significance of the work, I encourage the authors to engage with these empirical findings and clarify how their model aligns with, contrasts with, or extends these previous results.

Some relevant references that may be worth considering include:

Costello, F., & Watts, P. (2014). Surprisingly rational: Probability theory plus noise explains biases in judgment. Psychological Review, 121(3), 463–480.

Shirado, H., & Christakis, N. A. (2017). Locally noisy autonomous agents improve global human coordination in network experiments. Nature, 545, 370–374.

Sanborn, A. N., et al. (2025). Noise in cognition: Bug or feature? Perspectives on Psychological Science, 20(3), 572–589.

Addressing these studies and situating the current theoretical model within this broader empirical context would help clarify the paper’s contribution and increase its overall interpretive value for readers.

**Do you want your identity to be public for this peer review?** For information about this choice, including consent withdrawal, please see our Privacy Policy

Reviewer #1: No

Reviewer #2: No

---

## [Author Response · Author response to Decision Letter 1]

17 Nov 2025

Dear Reviewers and Editor,

Thank you for the thoughtful consideration and reviews of our manuscript, “When Noise Mitigates Bias in Human–Algorithm Decision-Making: An Agent-Based Model,” submitted for consideration as a Research Article in PLOS ONE.

Reviewer One provided two excellent points of feedback. The first concerned the number of “bad” individuals in the model simulations and why this does not influence model dynamics. We expanded on this in the manuscript on page 8, providing additional clarification on our decision to reduce complexity in order to isolate key mechanisms. Additionally, we offered guidance on page 18 regarding how future research could incorporate a trait-based approach.

The second point addressed the asymmetrical behavioral patterns observed in polarized groups. To strengthen this section, we added a new example concerning the politics of immigration, another highly polarized issue, that follows a similar pattern to the one established in our model.

Reviewer Two directed us to a critical body of literature relevant to our manuscript, including studies on the potential empirical benefits of noise in decision-making. Throughout the introduction and discussion, we incorporated this relevant scholarship. These additions improved the manuscript by situating our research within the emerging literature on the benefits of noise in decision-making.

We also addressed all journal and editorial requirements and suggestions mentioned in the review. These, along with all reviewer comments, were invaluable in improving the manuscript. Below, we detail our responses to each comment and suggestion, and indicate where in the manuscript we made the corresponding revisions. Please see the table summarizing our changes in the attached "Response to Reviewers" file.

Thank you again for your time and consideration. We look forward to the possibility of contributing this work to PLOS ONE.

Sincerely,

Spencer Poodiack Parsons

University of Twente

s.j.poodiack-parsons@utwente.nl

Tel: +31 681355961

Prof. dr. René Torenvlied

University of Twente

r.torenvlied@utwente.nl

Tel: +31 534892825

---

## [Decision Letter · Decision Letter 1]

7 Dec 2025

When noise mitigates bias in human–algorithm decision-making: An agent-based model

PONE-D-25-53741R1

Dear Dr. Poodiack Parsons,

We’re pleased to inform you that your manuscript has been judged scientifically suitable for publication and will be formally accepted for publication once it meets all outstanding technical requirements.

Kind regards,

Matteo Bodini, Ph.D.

Academic Editor

PLOS ONE

Additional Editor Comments (optional):

Reviewers' comments:

Reviewer's Responses to Questions

**Comments to the Author**

Reviewer #1: All comments have been addressed

Reviewer #2: All comments have been addressed

2. Is the manuscript technically sound, and do the data support the conclusions?

Reviewer #1: Yes

Reviewer #2: Yes

3. Has the statistical analysis been performed appropriately and rigorously?

Reviewer #1: Yes

Reviewer #2: Yes

4. Have the authors made all data underlying the findings in their manuscript fully available?

Reviewer #1: Yes

Reviewer #2: Yes

5. Is the manuscript presented in an intelligible fashion and written in standard English?

Reviewer #1: Yes

Reviewer #2: Yes

Reviewer #1: After this second round, I am now satisfied with the outcome and feel confident we can move forward without further revisions.

Reviewer #2: I appreciate the authors’ thorough efforts in revising the manuscript. I believe the paper is now ready for publication.

**Do you want your identity to be public for this peer review?** For information about this choice, including consent withdrawal, please see our Privacy Policy

Reviewer #1: No

Reviewer #2: No

---

## [Editor Report · Acceptance letter]

PONE-D-25-53741R1

PLOS One

Dear Dr. Poodiack Parsons,

I'm pleased to inform you that your manuscript has been deemed suitable for publication in PLOS One. Congratulations! Your manuscript is now being handed over to our production team.

Kind regards,

on behalf of

Dr. Matteo Bodini

Academic Editor

PLOS One